# Genetic Enhancement for Biotic Stress Resistance in Basmati Rice through Marker-Assisted Backcross Breeding

**DOI:** 10.3390/ijms242216081

**Published:** 2023-11-08

**Authors:** Gagandeep Singh, Niraj Singh, Ranjith Kumar Ellur, Alexander Balamurugan, G. Prakash, Rajeev Rathour, Kalyan Kumar Mondal, Prolay Kumar Bhowmick, S. Gopala Krishnan, Mariappan Nagarajan, Rakesh Seth, K. K. Vinod, Varsha Singh, Haritha Bollinedi, Ashok Kumar Singh

**Affiliations:** 1Division of Genetics, ICAR-Indian Agricultural Research Institute, New Delhi 110012, Indianiraj.singh37@gmail.com (N.S.); prolaybhowmick@gmail.com (P.K.B.); gopal_icar@yahoo.co.in (S.G.K.); kkvinodh@gmail.com (K.K.V.);; 2Division of Plant Pathology, ICAR-Indian Agricultural Research Institute, New Delhi 110012, Indiaprakashg.ganesan@gmail.com (G.P.);; 3Department of Agriculture Biotechnology, CSKHPKV, Palampur 176062, Himachal Pradesh, India; 4Rice Breeding and Genetics Research Centre, ICAR-Indian Agricultural Research Institute, Aduthurai 612101, Tamil Nadu, India; 5Regional Station, ICAR-Indian Agricultural Research Institute, Karnal 132001, Haryana, India; rseth101@gmail.com

**Keywords:** PB1509, bacterial blight, blast, SNP array, NILs

## Abstract

Pusa Basmati 1509 (PB1509) is one of the major foreign-exchange-earning varieties of Basmati rice; it is semi-dwarf and early maturing with exceptional cooking quality and strong aroma. However, it is highly susceptible to various biotic stresses including bacterial blight and blast. Therefore, bacterial blight resistance genes, namely, *xa13* + *Xa21* and *Xa38*, and fungal blast resistance genes *Pi9* + *Pib* and *Pita* were incorporated into the genetic background of recurrent parent (RP) PB1509 using donor parents, namely, Pusa Basmati 1718 (PB1718), Pusa 1927 (P1927), Pusa 1929 (P1929) and Tetep, respectively. Foreground selection was carried out with respective gene-linked markers, stringent phenotypic selection for recurrent parent phenotype, early generation background selection with Simple sequence repeat (SSR) markers, and background analysis at advanced generations with Rice Pan Genome Array comprising 80K SNPs. This has led to the development of Near isogenic lines (NILs), namely, Pusa 3037, Pusa 3054, Pusa 3060 and Pusa 3066 carrying genes *xa13* + *Xa21*, *Xa38*, *Pi9* + *Pib* and *Pita* with genomic similarity of 98.25%, 98.92%, 97.38% and 97.69%, respectively, as compared to the RP. Based on GGE-biplot analysis, Pusa 3037-1-44-3-164-20-249-2 carrying *xa13 + Xa21*, Pusa 3054-2-47-7-166-24-261-3 carrying *Xa38*, Pusa 3060-3-55-17-157-4-124-1 carrying *Pi9 + Pib*, and Pusa 3066-4-56-20-159-8-174-1 carrying *Pita* were identified to be relatively stable and better-performing individuals in the tested environments. Intercrossing between the best BC_3_F_1_s has led to the generation of Pusa 3122 (*xa13* + *Xa21* + *Xa38*), Pusa 3124 (*Xa38* + *Pi9* + *Pib*) and Pusa 3123 (*Pi9* + *Pib* + *Pita*) with agronomy, grain and cooking quality parameters at par with PB1509. Cultivation of such improved varieties will help farmers reduce the cost of cultivation with decreased pesticide use and improve productivity with ensured safety to consumers.

## 1. Introduction

Basmati rice is uniquely characterized by very-long cooked grains, delicious taste and exquisite aroma. It is a high-quality specialty rice from the Indo-Gangetic Plains, appreciated by consumers all over the world. The export of Indian Basmati rice earns more than US$ 4.7 billion annually [1]. However, this crop is constantly threatened by various biotic stresses of bacterial and fungal origin. Of them, bacterial blight (BB) disease caused by the *Xanthomonas oryzae pv. oryzae* (*Xoo*), a Gram-negative bacterium, and the rice blast caused by the heterothallic ascomycete fungal pathogen *Magnaporthe oryzae*, cause yield losses up to 50 to 90% [2]. These diseases are usually managed by the application of chemical pesticides. Pesticide residue in Basmati rice is one of the major concerns for the rejection of Basmati rice consignments. Furthermore, the application of chemical pesticides is not an ecologically friendly approach [3].

One of the leading Basmati rice varieties, Pusa Basmati 1509 (PB1509), with sturdy stem, semi-dwarf plant height (95–100 cm), early maturity (115–120 days), and non-shattering and exceptional grains [4], has contributed significantly to the financial reserves of the country through its export earnings. Further, due to early maturity, it provides enough opportunity to take up the timely sowing of subsequent crops. However, PB1509 is a potential target for incorporating genes for BB and blast resistance as it is highly susceptible to these diseases.

To date, 46 BB resistance genes have been mapped on various chromosomes, of which 28 genes are dominant and 18 genes are recessive in nature [5,6,7]. Among the mapped genes, eleven genes have been functionally characterized [8,9], and 15 *R*-genes were fine-mapped [6,7,10,11,12]. The BB resistance genes *Xa4*, *xa5*, *Xa7*, *xa13*, *Xa21* and *Xa38* were most frequently used in marker-assisted backcross breeding (MABB) programs for developing BB-resistant cultivars [13,14]. The BB resistance genes *Xa8*, *xa13*, *Xa21* and *Xa38* were effective for the isolates prevalent in the Basmati GI region [13].

The BB resistance gene *xa13* was identified in the genotype BJ1 and mapped onto the chromosome [15]. This gene interacts strongly with other resistance genes, namely, *Xa4*, *xa5* and *Xa21* [16]. Studies based on the pathogenicity analysis have revealed that *xa13* activates unique defense genes as compared to other major dominant *R* genes, namely *Xa4*, *Xa10* and *Xa26*, suggesting a unique resistance mechanism of the gene [17]. Another important broad-spectrum BB resistance gene *Xa21* was discovered in an accession of *Oryza longistaminata*. The *Xa21* gene was mapped to the long arm of chromosome 11 and a functional marker, pTA248 was developed for its effective deployment through marker-assisted selection [18]. Another dominant BB resistance gene *Xa38*, located on the long arm of chromosome 4, was identified from an accession of *Oryza nivara*, and a tightly linked InDel marker (marker-LOC_Os04g53050-1) co-segregating with the BB resistance was developed [19,20]. Moreover, pyramiding two or more BB resistance genes is considered effective as it increases the spectrum of resistance and reduces the speed of evolution of isolates virulent against the deployed genes. A combination of *xa13* and *Xa21* has been widely utilized in rice breeding programs and several varieties have been released [21,22]. The importance of another BB resistance gene, *Xa38*, in rice breeding programs, is highlighted by its utilization in the development of several BB-resistant genotypes [13,23]. Considering the resistant spectrum of *xa13*, *Xa21* and *Xa38*, the present study was targeted to introgress them into the genetic background of PB1509.

Another profoundly serious and devastating disease of rice is blast. Yield losses due to blast disease are variable depending on the genotype and environment, reaching up to 100% under disease-friendly conditions [24]. Similar to BB, the development of host plant resistance is the most effective strategy for the management of blast disease [25]. So far, more than 100 resistance genes have been identified and 37 of them have been cloned [26]. Although several blast resistance genes have been identified, only a few of them were used in breeding programs for blast disease management in India [27]. The major blast resistance gene, *Pi9*, mapped to the *Piz* locus on chromosome 6 confers resistance to 43 *M. oryzae* isolates prevalent in 13 countries. Of the six NBS-LRR domains present in this locus, Nbs2-*Pi9* is the functional blast resistance gene *Pi9* [28]. The *Pib* gene in rice is known to confer resistance to a wide range of races of the rice blast pathogen, including race IE1k [29]. Another blast resistance gene *Pita* located on chromosome 12 adjacent to the centromere in the genotype ‘Tetep’ [30] was found to encode a cytoplasmic protein-containing NBS-LRR domain [31,32]. The pyramiding of *Pi9*, *Pib* and *Pita* is a promising strategy to widen the resistant profile of genotypes as well as to achieve substantial resistance. There have been several reports of the successful incorporation of blast resistance genes into the genetic background of different rice varieties [33,34].

Considering the prevailing diseases in the Basmati growing regions of the country, developing resistance to both BB and blast diseases can prove to be an effective strategy as it can significantly reduce the use of chemical pesticides while reducing the cost of cultivation. Therefore, we attempted to incorporate BB (*xa13*, *Xa21* and *Xa38*) and blast (*Pi9*, *Pib* and *Pita*) resistance genes into the genetic background of PB1509 through MABB.

## 2. Results

### 2.1. Generation of PB1509-NILs Carrying BB and Blast Resistance Genes

For the development of NILs with resistance to biotic stresses, four donor parents (DPs), namely, PB1718, P1927, P1929 and Tetep, were crossed with the RP (PB 1509) to generate F_1_s, which were designated as Pusa 3037, Pusa 3054, Pusa 3060 and Pusa 3066, respectively. The list of polymorphic markers obtained in all the cross combinations is presented in Appendix A.

The BC_1_F_1_ plants generated from the cross PB1509/PB1718 carrying the target genes *xa13* and *Xa21* in a heterozygous state were identified through foreground selection using the markers xa13prom and pTA248, respectively, and the recurrent parent genome (RPG) recovery estimated using 74 polymorphic SSR markers ranged from 64.8% to 84.2% (Table 1). Similarly, in BC_1_F_1_s from the cross PB1509/P1927, foreground selection with marker-Os04g53050 led to the identification of plants carrying *Xa38* in a heterozygous state and RPG recovery estimated using 83 polymorphic SSR markers ranged from 73.8% to 88.6%. In the BC_1_F_1_s from the cross PB1509/P1929, the plants carrying *Pi9* and *Pib* were found to have RPG recovery of 70.7% to 81.0% with 84 polymorphic SSR markers. In the case of PB1509/Tetep, the BC_1_F_1_s carrying *Pita* in a heterozygous state had RPG recovery ranging from 57.3 to 87.5%. Further, based on RPG and RP phenotype recovery, the best plants in each of the cross combinations were identified for backcrossing to generate BC_2_F_1_s.

During BC_2_F_1_ generation, background selection with the remaining polymorphic markers and the markers that were heterozygous in the BC_1_F_1_ generation revealed RPG recovery of up to 94.4%, 93.3%, 91.9% and 86.4% in the cross combinations PB1509/P1927, PB1509/ PB1718, PB1509/P1929 and PB1509/Tetep, respectively. In each of the combinations, a BC_2_F_1_ plant with maximum resemblance to the phenotype of RP coupled with maximum RPG recovery was selected and backcrossed to generate BC_3_F_1_ seeds. The best BC_3_F_1_ plant was selfed to generate BC_3_F_2_ populations. A total of 12, 15, 14 and 15 plants homozygous for Pusa 3037, Pusa 3054, Pusa 3060 and Pusa 3066, respectively, were identified through foreground selection using gene-linked markers. Stringent phenotypic screening for RP phenotype was carried out till BC_3_F_5_ generation. A total of four NILs for Pusa 3037 and 5 NILs each for Pusa 3054, Pusa 3060 and Pusa 3066 were identified and subjected to multi-location evaluation for yield and its component traits, grain and cooking quality parameters.

### 2.2. Background Analysis of NILs

The NILs were genotyped using Rice Pan Genome Array (RPGA) (Figure 1). Among the PB1509-NILs carrying blast resistance genes *Pi9* + *Pib* and *Pita*, Pusa 3060-3-55-17-157-4-138-3 (G5) and Pusa 3066-4-56-20-158-7-170-2 (G6) possessed maximum genomic resemblance of 97.38% and 97.69%, respectively, with the RP. Of the NILs carrying BB resistance gene *Xa38*, a maximum genomic similarity of 98.92% was observed in the NIL Pusa 3054-2-47-7-166-24-261-2 (G16). Among the NILs carrying bacterial blight resistance genes *xa13* + *Xa21*, Pusa 3037-1-44-3-164-21-255-1 (G12) had a maximum similarity of 98.25% with the RP genome.

### 2.3. Agronomic, Grain and Cooking Quality Parameters of NILs

The agro-morphological, grain and cooking quality revealed the on-par performance of NILs with the RP but for some exceptions (Table 2). Days to fifty percent flowering (DF) is the vital parameter that determines the maturity duration of the crop. DF was significantly lower in Pusa 3066-4-56-20-159-8-174-1 (G10) (81.0 days), Pusa 3060-3-55-17-157-4-138-1 (G3) (83.3 days) and Pusa 3060-3-55-17-157-4-138-2 (G4) (83.0 days) as compared to PB1509 (85.0 days). Pusa 3060-3-55-17-157-4-138-2 (G4) and Pusa 3054-2-47-7-166-24-261-2 (G16) possessed significantly higher test weight (TW) as compared to RP.

The NILs were either at par or significantly superior to RP (Table 3; Figure 2). Pusa 3066-4-56-20-158-7-172-1 (G8) possessed significantly higher hulling recovery (HR); the NILs Pusa 3066-4-56-20-158-7-171-4 (G7), Pusa 3037-1-44-3-164-21-255-1 (G12), Pusa 3037-1-44-3-164-21-256-4 (G13), Pusa 3037-1-45-5-165-22-259-4 (G14), Pusa 3054-2-47-7-166-24-261-1 (G15) and Pusa 3054-2-47-7-166-24-262-3 (G19) possessed higher milling recovery (MR) as compared to the RP. Another important quality parameter, Kernel length before cooking (KLBC), was significantly higher for the NILs, Pusa 3037-1-45-5-165-22-259-4 (G14), Pusa 3054-2-47-7-166-24-261-1 (G15) and Pusa 3054-2-47-7-166-24-262-3 (G19). The NIL Pusa 3054-2-47-7-166-24-262-2 (G18) possessed significantly higher Kernel length after cooking (KLAC). All the NILs exhibited an elongation ratio (ER) of more than two except Pusa 3060-3-55-17-157-4-138-2 (G4).

### 2.4. Multi-Location Evaluation

A set of 19 NILs were evaluated under three environments, viz., New Delhi (Env1), Karnal (Env2) and Urlana (Env3), in randomized complete block design (RCBD) with three replications during *kharif* 2021. The yield data for NILs evaluated at three locations are presented in Table 4. Three NILs for *xa13 + Xa21*, Pusa 3037-1-44-3-164-20-249-2 (G11), Pusa 3037-1-44-3-164-21-255-1 (G12) and Pusa 3037-1-44-3-164-21-256-4 (G13) were significantly higher yielding under the conditions of Env1. Among the NILs carrying *Xa38*, Pusa 3054-2-47-7-166-24-261-1 (G15), Pusa 3054-2-47-7-166-24-261-2 (G16) and Pusa 3054-2-47-7-166-24-262-2 (G18) significantly outyielded RP in Env1. NILs carrying blast resistance gene *Pita*, viz., Pusa 3066-4-56-20-158-7-171-4 (G7) and Pusa 3066-4-56-20-158-7-172-2 (G9), yielded significantly higher as compared to the RP. In Env2, all the NILs performed on par except for Pusa 3060-3-55-17-157-4-138-2 (G4), which was inferior to RP. Under Env3, all the NILs were on par with RP, except Pusa 3060-3-55-17-157-4-138-1, Pusa 3060-3-55-17-157-4-138-2 and Pusa 3060-3-55-17-157-4-138-3 carrying blast resistance genes *Pi9 + Pib* and Pusa 3054-2-47-7-166-24-262-3 carrying *Xa38*, which significantly underperformed.

Genotype ranking biplot detects ideal genotypes in comparison to the other evaluated genotypes. The cumulative variation explained by two components in a GGE biplot was 91.19%. NILs carrying BB resistance genes *xa13* + *Xa21*, namely, Pusa 3037-1-44-3-164-20-249-2 (G11) and Pusa 3037-1-44-3-164-21-256-4 (G13); and the NILs carrying *Xa38*, viz., Pusa 3054-2-47-7-166-24-261-2 (G16), Pusa 3054-2-47-7-166-24-261-3 (G17) and Pusa 3054-2-47-7-166-24-262-2 (G18), could be considered as the best-performing genotypes owing to their presence in the innermost circle (Figure 3). The genotypes located in the inner circle are highly desirable as compared to the ones in the outer circle. Among the NILs with blast resistance genes *Pi9 + Pib*, Pusa 3060-3-55-17-157-4-124-1 (G1) was found to be desirable owing to its presence in the circle next to the innermost circle (Figure 3).

In the polygon pattern plot, the environmental indicators were positioned into two segments with different genotypes winning in each segment. The biplot was divided into six clockwise fan-shaped sections. Two NILs, Pusa 3037-1-44-3-164-21-255-1 (G12) and Pusa 3037-1-44-3-164-20-249-2 (G11) were high yielding and stable in Env1 and Env3, whereas one NIL, Pusa 3054-2-47-7-166-24-261-3 (G17), outperformed other genotypes and was found to be highly stable in Env2. The mean vs. stability analysis plot (Figure 4) identified Pusa 3054-2-47-7-166-24-261-2 (G16) followed by Pusa 3037-1-44-3-164-21-256-4 (G13) and Pusa 3054-2-47-7-166-24-262-2 (G18) to be high performing in Env1 and Env3, while in case of Env2, Pusa 3037-1-44-3-164-20-249-2 (G11) was the highest performer followed by Pusa 3054-2-47-7-166-24-261-3 (G17). Apart from these NILs, Pusa 3037-1-44-3-164-21-255-1 (G12) and Pusa 3060-3-55-17-157-4-124-1 (G1) provided acceptable yield but low stability because of their position away from the Average Environment Coordinate (AEC) line. We recorded Env1 and Env2 as independent and unique locations for the yield due to their shorter vector length. However, the shorter-angled-lengthy environment vector with the AEC line is ideal for the selection of suitable genotypes among the total genotypes. Thus, the greater representation and discrimination are indicated by the test environment, Env3.

### 2.5. Combining Genes Governing Multiple Disease Resistance

Simultaneously, the best BC_3_F_1_ plant with maximum RPG and RP phenotype recovery were intercrossed to generate Pusa 3122 (Pusa 3037/Pusa 3054), Pusa 3123 (Pusa 3037/Pusa 3060) and Pusa 3124 (Pusa 3054/Pusa 3060), carrying *xa13* + *Xa21* + *Xa38*, *Pi9* + *Pib* + *Pita* and *Xa38* + *Pi9* + *Pib*, respectively. All the NILs developed were on par with PB1509 for agronomic, yield and yield-related traits (Table 5). DF was significantly lower (82.5 days) in the NIL for *Pi9 + Pib + Pita* and Pusa 3123-33-13-312-25 (G29) in comparison to RP (85.5 days). The trait affecting the yield of the crop, NT, was significantly higher in Pusa 3124-38-19-176-5 (G24) with an average of 20.5 productive tillers/plant. In terms of their grain and cooking quality, all the NILs exhibited grain parameters similar to PB1509 and had an excellent cooking quality with an elongation ratio of greater than two (Table 6 and Figure 4).

### 2.6. Screening for BB and Blast Resistance

During *kharif* 2020 and 2021, the NILs were evaluated for BB resistance using the *Xoo* race 4 (Appendix A, Figure 5 and Figure 6). The RP was highly susceptible with lesion lengths of 14.37 ± 0.71 cm, while the DPs, PB1718 (*xa13* + *Xa21*) and P1927 (*Xa38)*, were highly resistant with lesion lengths of 1.77 ± 0.45 cm and 0.61 ± 0.17 cm, respectively. The NILs carrying *xa13 + Xa21* showed highly resistant reactions with the lesion lengths ranging from 1.68 ± 0.44 cm in the NIL, Pusa 3037-1-45-5-165-22-259-4 (G14) to 3.49 ± 0.80 cm in Pusa 3037-1-44-3-164-20-249-2 (G11). The *Xa38*-carrying NILs possessed lesion lengths ranging from 0.33 ± 0.06 cm in Pusa 3054-2-47-7-166-24-262-2 (G18) to 2.76 ± 0.80 cm in Pusa 3054-2-47-7-166-24-261-3 (G17). In the NILs carrying *xa13* + *Xa21* + *Xa38*, the lesion length ranged from 0.25 ± 0.05 cm in Pusa 3122-27-16-166-1 (G21) to 0.49 ± 0.11 cm in Pusa 3122-27-15-165-2 (G20). In the NILs carrying *Xa38* + *Pi9* + *Pib* for bacterial blight resistance, lesion lengths ranged from 0.21 ± 0.008 cm in Pusa 3124-40-21-179-4 (G25) to 0.43 ± 0.09 cm in Pusa 3124-40-22-180-2 (G26).

Screening for the blast resistance was carried out at IARI, New Delhi, and CSKHPKV, Palampur, for two seasons. All the NILs carrying blast resistance genes *Pi9 + Pib* and *Pita* exhibited resistance reactions for the isolate *Mo-ei-MB20* under artificial epiphytotics at New Delhi (Table 7). The PB1509-NILs carrying *Pi9 + Pib*, namely, Pusa 3060-3-55-17-157-4-124-1 (G1), Pusa 3060-3-55-17-157-4-124-6 (G2), Pusa 3060-3-55-17-157-4-138-1 (G3), Pusa 3060-3-55-17-157-4-138-3 (G5) and *Pita*-carrying NILs, viz., Pusa 3066-4-56-20-158-7-171-4 (G7) and Pusa 3066-4-56-20-158-7-172-2 (G9), produced a resistant reaction with the disease score of one. The NILs carrying *Pi9 + Pib + Pita* were highly resistant with a disease score of zero. Among the NILs carrying *Xa38* + *Pi9* + *Pib*, viz., Pusa 3124-38-19-176-5 (G24) and Pusa 3124-40-22-180-2 (G26), maximum resistance was shown to the blast isolate with a disease score of one. With the mixture of five races, viz., *Po-RML21*, *Po-RML29*, *Po-HP5-2*, *Po-NWI-102* and *Po-NWI-141*, all the NILs tested were found to be resistant.

## 3. Discussion

PB1509 is an elite Basmati rice variety and is popular among farmers owing to its early maturity, semi-dwarfness, exquisite grain and cooking qualities. Furthermore, it contributes immensely to the national exchequer due to its export potential. However, it suffers adversely due to biotic stresses imposed by disease-causing organisms such as *Xoo* and *M. oryzae*. Choosing pesticides wisely and their timely application is essential for the management of this disease. However, there are several concerns associated with environmental hazards and consumer safety. Additionally, the rise in the stringency of pesticide residue limits by importing nations has led to concerns of the rejection of Basmati rice consignments. Therefore, developing inbuilt genetic resistance is the only viable option for the maintenance of healthy international trade, the environment, and satisfied consumers. MABB is considered an effective approach for arming the Basmati rice cultivars with inbuilt disease resistance [35]. To ameliorate the issue of susceptibility to diseases, the present study was aimed at incorporating three bacterial blight resistance genes, namely, *xa13*, *Xa21* and *Xa38*, and blast resistance genes, namely, *Pi9*, *Pib* and *Pita*, in various combinations, into the genetic background of PB1509.

The *Xoo* pathogen is known to secrete transcription-activator-like effectors (TALEs), which target promoters for induction of one of the *SWEET* genes (*SWEET11*, *SWEET13* and *SWEET14*). The major BB resistance gene *xa13* provides race-specific resistance and it is also referred to as *SWEET11*. The other dominant gene *Xa21* is known to encode a receptor kinase with the NBS-LRR domain, which recognizes a conserved determinant present in multiple races of *Xoo* [36]. The combination of *xa13* and *Xa21* has been proven to provide synergistic action in imparting elevated levels of resistance to the *Xoo* isolates. Therefore, *xa13 + Xa21* has been widely deployed into rice varieties such as Pusa Basmati 1, Ranbir Basmati, PRR78, Pusa 6B, etc. [37,38]. However, recently there have been reports of the breakdown of resistance governed by *xa13 + Xa21* [39]. Subsequently, a major dominant BB resistance gene *Xa38* with broad-spectrum resistance mapped on chromosome 4 [19] was incorporated into a series of elite rice varieties, viz., Pusa Basmati 1121, Super Basmati and Improved Samba Mahsuri [13,23,40]. The NILs carrying *Xa38* were resistant to a greater number of races of *Xoo* as compared to the NILs carrying *xa13 + Xa21*, indicating a wider spectrum of resistance governed by the gene *Xa38* [13]. The pyramiding of multiple genes improves the resistance spectrum of a genotype as well as reduces the speed of evolution of pathogens. Therefore, incorporating *xa13* + *Xa21* and *Xa38* into the genetic background of PB1509 was considered a better choice to have prolonged resistance against multiple races of *Xoo*. We used PB1718 and P1927, carrying BB resistance genes *xa13 + Xa21* and *Xa38*, respectively, as DPs.

For blast resistance, *Pi9* and *Pita* were identified as effective against the *M. oryzae* isolates prevalent in the Basmati rice growing region of the country, which includes the states of Punjab, Haryana, Delhi, Jammu and Kashmir, Himachal Pradesh, Western Uttar Pradesh and Uttarakhand [41,42,43]. Therefore, Pusa 1929 carrying *Pi9* and *Pib* in the genetic background of Pusa Basmati 1, and Tetep carrying *Pita*, was used as DPs. The successful incorporation of blast resistance genes, namely, *Pi9*, *Pi2*, *Pi54*, *Pita*, *Pib* and *Pi1* in Basmati varieties, has been demonstrated earlier [41]. Several blast-resistant varieties, namely, Pusa Basmati 1609, Pusa Basmati 1637, Pusa 1612, Pusa Basmati 1847, etc., have been developed and released for commercial cultivation [34,44].

Marker-assisted backcross breeding is the effective approach for defect correction in the otherwise agronomically superior varieties. Three rounds of backcrossing were employed to ensure sufficient recovery of the RP genome. In each of the backcross generations, foreground selection ensured the incorporation of the target gene, and background and phenotypic selection identified the plant with maximum recovery for the RP genome and phenotype, respectively. The current strategy has led to the development of five NILs carrying *Xa38* (Pusa 3054) and four NILs carrying *xa13 + Xa21* (Pusa 3037), which exhibited an RP genome similarity of more than 97%. The NILs carrying *xa13 + Xa21* were highly resistant with BB lesion lengths ≤ 3.5 cm, while the NILs carrying *Xa38* possessed lesion lengths of ≤3.0 cm. The NILs carrying *xa13* + *Xa21* + *Xa38* possessed lesion lengths of ≤1.0 cm. This indicates that *xa13 + Xa21*, *Xa38* and *xa13 + Xa21 + Xa38* were effective in conferring resistance against the virulent isolates dominant in Basmati-producing regions of the country. The successful introgression of bacterial blight resistance genes *xa13*, *Xa21*, *Xa4*, *xa5*, *Xa33*, *Xa40*, etc., has been earlier reported [14,45,46,47]. Similarly, five NILs both for *Pi9* + *Pib* (Pusa 3060) and *Pita* (Pusa 3066) were developed, which were highly resistant to the prevalent isolates *Mo-ei-MB20*, *Po-RML21*, *Po-RML29*, *Po-HP5-2*, *Po-NWI-102* and *Po-NWI-14* under artificial inoculated conditions.

Background selection using SSR markers during early backcross generations has led to maximizing RPG recovery, which ranged from 86.1% to 99.6% during BC_3_F_1_ generation. SSR markers were cost-effective, accessible, and with better ease of handling in the laboratory [48]. Therefore, SSRs were suited for background selection during backcross generations. Further, high-density SNP assays aid in having accurate estimates of background recovery as well as DP introgressions [49]. Therefore, RPGA comprising 80K SNPs was used for background analysis in the advanced generations for identifying the lines with maximum similarity to RP [38]. The effectiveness of the approach was evident from the genome similarity values of more than 96% observed in NILs as compared to RP, with a maximum similarity of 98.92% in the NIL, Pusa 3054-2-47-7-166-24-261-2 (G16).

An ideal genotype should possess better stability with high per se performance. Therefore, multi-environment testing of NILs was performed. In ranking genotypes of GGE biplot, the ring at the head of the arrow on the horizontal axis represents an ideal genotype [50]. The NILs, Pusa 3037-1-44-3-164-20-249-2 (G11), Pusa 3037-1-44-3-164-21-256-4 (G13), Pusa 3054-2-47-7-166-24-261-2 (G16), Pusa 3054-2-47-7-166-24-261-3 (G17) and Pusa 3054-2-47-7-166-24-262-2 (G18) were identified as near ideal genotypes for yield per hectare as they were placed in the innermost circle near to the hypothetical ideal genotype.

Considering the mean vs. stability analysis plot, the arrow sign on the AEC line ranks the genotypes in increasing order. The NILs, G16, G13 and G18 were high-yielding in Env1 and Env3, while G11 and G17 were the genotypes with higher yield in Env2. Pusa 3037-1-44-3-164-21-255-1 (G12) and Pusa 3060-3-55-17-157-4-124-1 (G1) were less stable but with a high yield. Therefore, this genotype can be a potential line for the crop improvement program. A genotype placed on the horizontal axis with zero vertical projection is considered more stable, while a genotype with a lengthy direction from the AEC abscissa is treated as an unstable genotype [51]. In the ‘which-won-where’ pattern plot, the environmental indicators are positioned into two segments, confirming the presence of distinct interaction between genotype and environment. The genotype that is attached to the vertex of polygon where the environmental marker drops suggests that the genotype gives a high yield and performs best in that environment [52]. Therefore, G11 was the best performer in Env 1 and 3, whereas G1 was best for Env 2.

Several rice varieties with resistance to either bacterial blight or blast have been developed and released for cultivation, which provides farmers an opportunity to choose according to the disease prevalence in their growing environment. However, in the majority of the Basmati growing regions, BB and blast co-exist, which raises the need for combining bacterial blight and blast resistance into high-yielding varieties [53]. Therefore, PB1509 NILs carrying *xa13 + Xa21 + Xa38*, *Pi9 + Pib + Pita* and *Xa38 + Pi9 + Pib* were developed. These lines were effective against the BB and blast isolates prevalent in the Basmati growing regions of the country. The adoption of multi-disease resistant NILs would reduce the application of chemical pesticides, paving the way for eco-friendly rice cultivation, which is safe for consumers, while strengthening the international trade of Basmati.

## 4. Materials and Methods

### 4.1. Plant Material

PB1509, an elite Basmati rice variety was used as RP; PB1718 carrying BB resistance genes *xa13 +Xa21*, P1927 carrying BB resistance gene *Xa38*, P1929 carrying blast resistance genes *Pi9 + Pib*, and Tetep possessing blast resistance gene *Pita* were used as the DPs.

### 4.2. Breeding Strategy

The breeding methodology adopted is presented in Figure 7. RP was crossed with four different DPs to generate F_1_ seeds. The F_1_ plant was selected based on a test of hybridity and backcrossed with RP to produce BC_1_F_1_ seeds. Foreground selection was then carried out on BC_1_F_1_ plants to identify the heterozygous plants, which were then subjected to background selection followed by phenotypic selection for agro-morphological and grain quality parameters. A BC_1_F_1_ plant with maximum resemblance to recurrent parent was backcrossed to generate BC_2_F_1_ seeds. Subsequently, same strategy was followed to generate BC_3_F_1_ seeds. The BC_3_F_1_ plant with superiority was selfed, and the plants homozygous for genes of interest were identified in BC_3_F_2_ populations. These lines were tested for grain and cooking quality traits and advanced through pedigree selection till BC_3_F_5_ generation.

Simultaneously, intercrosses were made between the best BC_3_F_1_s plants to generate intercross F_1_s carrying *xa13* + *Xa21* + *Xa38*, *Xa38* + *Pi9* + *Pib* and *Pi9* + *Pib* + *Pita*. The plants homozygous for the respective genes were identified in the intercross F_2_ populations and were advanced till F_6_ generation through pedigree selection. The developed NILs were evaluated before being nominated into the national system of varietal release.

### 4.3. DNA Extraction and PCR

The genomic DNA was extracted from 15-day-old seedlings using CTAB buffer with slight modification to the protocol [54]. PCR of 10 μL volume was set up using 20–30 ng template DNA, 5 pmol of each primer and EmeraldAmp Max PCR Master Mix (2X Premix, Takara) using the Biorad T100TM thermal cycler with the standard PCR program and electrophoresis was carried out using Metaphor™ agarose gel and visualized on Gel DocTM XR+ documentation System.

### 4.4. Foreground Selection

Foreground selection for the genes *xa13*, *Xa21* and *Pi9* was carried out using the gene-based markers, namely, xa13prom, pTA248 and Pi9STS1, respectively. The selection for the genes *Xa38*, *Pib* and *Pita* was conducted using gene-linked markers, viz., marker-Os04g53050-1, RM535 and RRS12, respectively.

### 4.5. Recurrent Parent Genome Recovery Using SSR and SNP Markers

The RP and the DPs were surveyed for polymorphic markers using 735 genome-wide SSR markers from the URL link http://www.gramene.org (accessed on 22 August 2016) and the primers were synthesized. A final set of developed NILs were subjected to estimation of RPG similarity using Affymetrix based 80K RPGA [55]. The genome similarity was visualized using the Phenogram software from the URL link http://visualization.ritchielab.psu.edu/phenograms/plot (accessed on 12 February 2023). The similarity to the genome of RP was obtained using the formula, (R + 1/2H) X 100/P, where R = Number of markers amplifying homozygous allele for recurrent parent, H = number of heterozygous markers and P = total number of markers.

### 4.6. Screening for Disease Resistance

For evaluation of BB resistance, the parental lines and NILs were grown in the field conditions. The suspension of *Xoo* race 4 with a density of 10^9^ cells/mL was prepared. Inoculation was performed with the isolate by clip inoculation method wherein top five leaves from every entry were clipped [56]. The length of BB lesion was measured post-21 days of inoculation. The genotype with a lesion length of up to 5 cm was considered to be resistant, 5–10 cm was considered to be moderately resistant reaction, 10–15 cm was considered to be moderately susceptible response, and more than 15 cm to be as completely susceptible [57].

To screen for blast resistance, the NILs carrying blast resistance genes along with the RP and DPs were grown in pro-trays as per the protocol [58]. The seedlings were grown under optimum conditions at temperature of 28 °C and 95% relative humidity. At three-leaf stage, the inoculum of *M. oryzae* isolate comprising of approximately 5 × 10^4^ conidia per ml was mixed with 0.02% tween 20 and sprayed on the seedlings. At IARI, New Delhi, *M. oryzae* isolate *Mo-ei-MB20* was used for screening. Mixture of five isolates, namely, *Po-RML21*, *Po-RML29*, *Po-HP5-2*, *Po-NWI-102* and *Po-NWI-141*, constituted the inoculum for screening at Palampur. The seedlings were scored for the blast resistance 7 days post-inoculation following Bonman’s scale. Scores of 0–3 were considered resistant reactions, and scores of 4 and 5 were considered susceptible reactions.

### 4.7. Multi-Environment Agro-Morphological Evaluation of Developed NILs

The NILs along with the RP and DPs were evaluated for agronomic performance with three replications in RCBD at three different environments, namely, IARI-New Delhi (Env1), IARI Regional Station-Karnal (Env2), and Urlana (Env3), during *kharif* 2021 following recommended agronomic practices. Data for traits, viz., DF, PH, PL, NT, NFG, SF, TW and grain yield, were recorded. DF was recorded on a plot basis. A representative ten plants from each of the NILs were considered for measurement of other traits.

### 4.8. Grain and Cooking Quality Evaluation of Developed NILs

Traits related to quality, namely, HR, MR, HRR, KLBC, KBBC, KLAC, KBAC, ER, and aroma, were recorded following standard protocol. HR was calculated in percentage as the ratio of the weight of whole polished grains to the weight of the raw grains. Grain parameters such as KLBC and KBBC were recorded on a photo enlarger using ten grains from each of the NILs. For the recording of cooking quality characteristics, namely, KLAC and KBAC, ten whole milled kernels were selected and soaked for 30 min in 10 mL of distilled water taken in test tubes. The lower part of the tubes with rice kernels was then immersed in a boiling water bath for 8–10 min. The cooked kernels were cooled to room temperature after transferring the contents into a Petri plate, and data were recorded. The data were subjected to statistical analysis using the package CropStat 7.2 [59], and stability analysis was carried out using the Metan package v 1.18.0 [60].

## Figures and Tables

**Figure 1 ijms-24-16081-f001:**
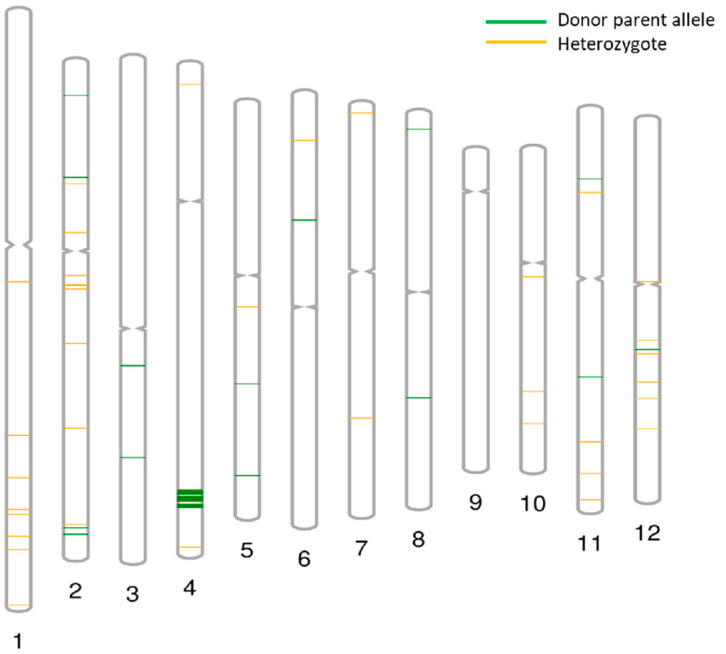
Representation of the recurrent parent similarity for Pusa 3054-2-47-7-166-24-261-3 (G17). 1–12 represents the chromosome numbers.

**Figure 2 ijms-24-16081-f002:**
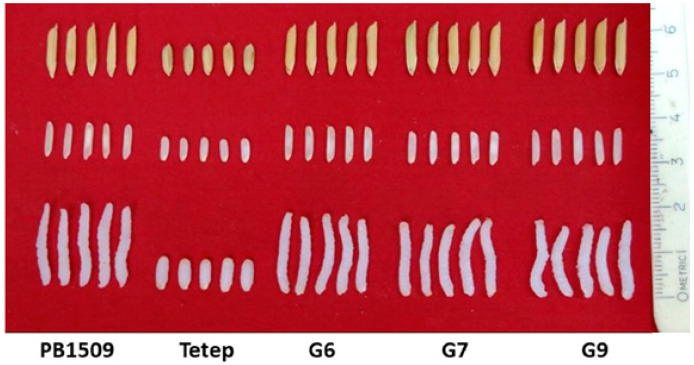
Grain and cooking quality of the parental lines and the NILs carrying *Pita* where G6 = Pusa 3066-4-56-20-158-7-170-2, G7 = Pusa 3066-4-56-20-158-7-171-4 and G9 = Pusa 3066-4-56-20-158-7-172-2.

**Figure 3 ijms-24-16081-f003:**
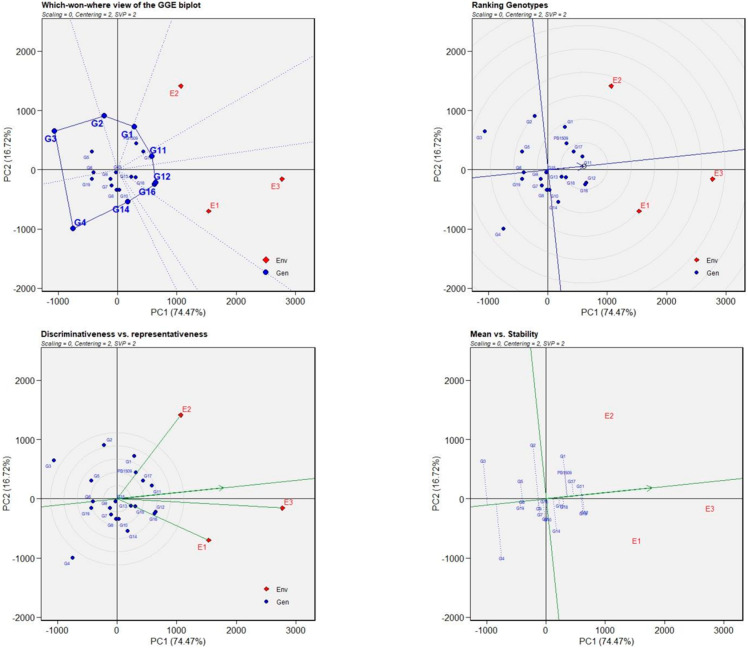
Genotype main effect (G) plus genotype by environment interaction (GE) biplot for the yield data of NILs evaluated during *kharif* 2021 at three different locations. G1 to G5 PB1509 + *Pi9* + *Pib*, G6 to G10-PB1509 + *Pita*, G11 to G14 -PB1509 + *xa13* + *Xa21* and G15 to G19-PB1509 + *Xa38*.

**Figure 4 ijms-24-16081-f004:**
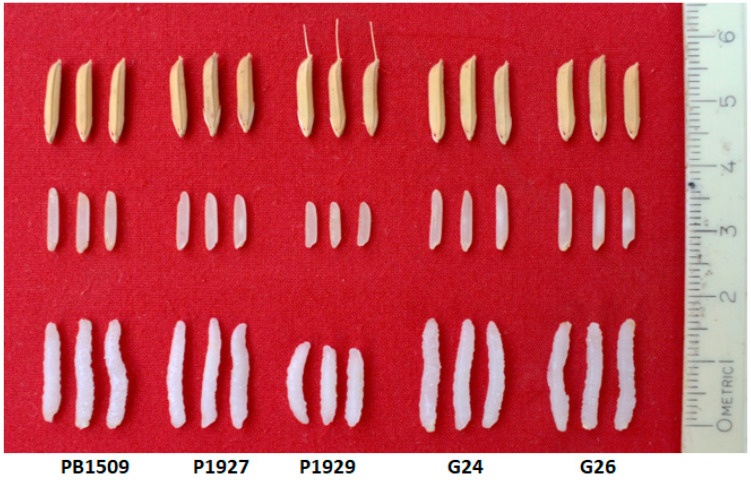
Grain and cooking quality of the parental lines and the NILs carrying *Xa38*, *Pi9* and *Pib* where G24 = Pusa 3124-38-19-176-5 and G26 = Pusa 3124-40-22-180-2.

**Figure 5 ijms-24-16081-f005:**
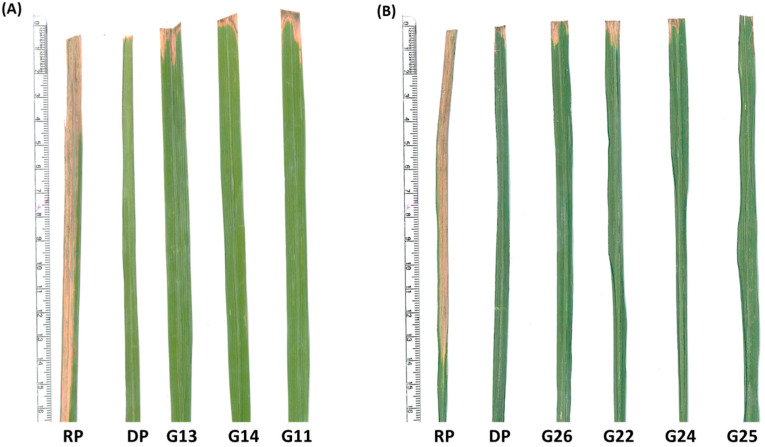
BB disease reaction under artificial inoculation conditions: (**A**) RP- PB1509, DP-PB1718, G13-Pusa 3037-1-44-3-164-21-256-4, G14-Pusa 3037-1-45-5-165-22-259-4 and G11-Pusa 3037-1-44-3-164-20-249-2. (**B**) RP-PB1509, DP-P1927, G26-Pusa 3124-40-22-180-2, G22-Pusa 3124-37-17-168-4, G24-Pusa 3124-38-19-176-5 and G25-Pusa 3124-40-21-179-4.

**Figure 6 ijms-24-16081-f006:**
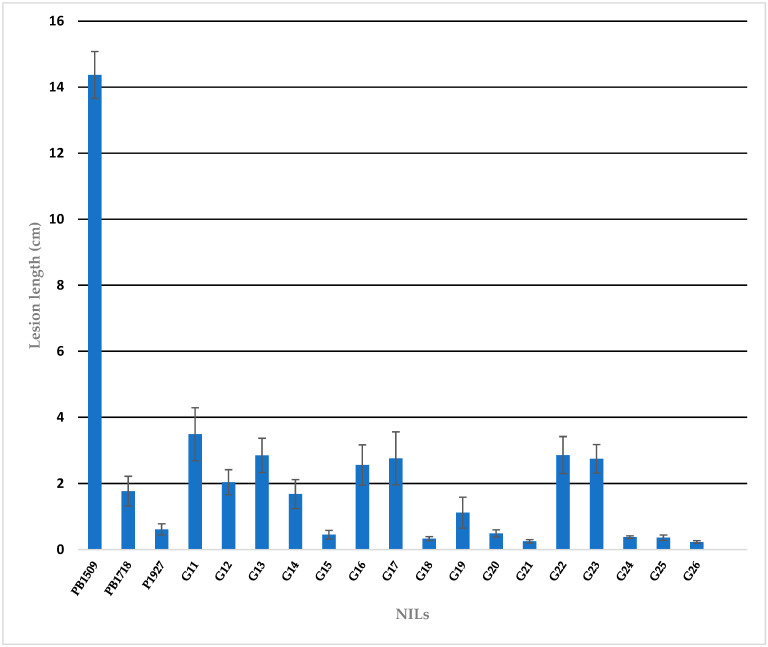
The bacterial blight lesion length in NILs and parental lines. The values represented are the mean of replicated data over two seasons and the standard error is shown as error bars.

**Figure 7 ijms-24-16081-f007:**
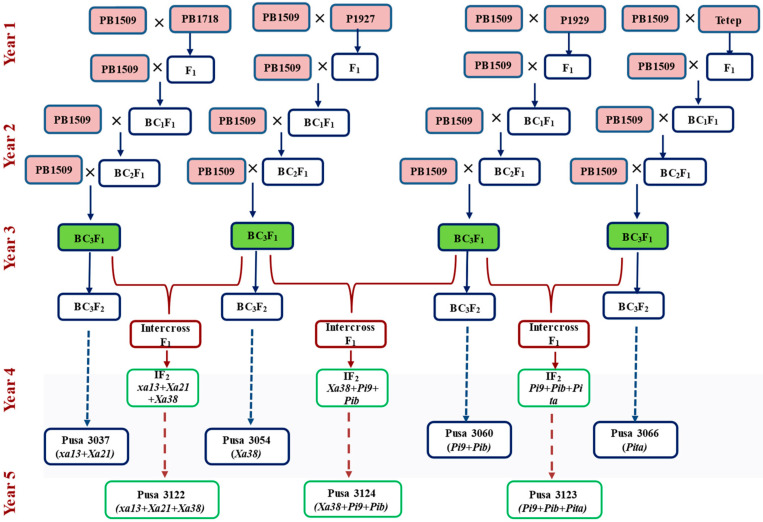
MABB scheme for development of NILs.

**Table 1 ijms-24-16081-t001:** Recovery of recurrent parent genome (RPG) during different stages of backcross breeding using polymorphic SSR markers. Pusa 3037-PB1509 + *xa13* + *Xa21*; Pusa 3054-PB1509 + *Xa38*; Pusa 3060-PB1509 + *Pi9* + *Pib*; Pusa 3066-PB1509 + *Pita*.

Genotype	RPG Recovery (%)		
	BC_1_F_1_	BC_2_F_1_	BC_3_F_1_
Pusa 3037	64.8–84.2	83.3–93.3	86.1–99.6
Pusa 3054	73.8–88.6	91.7–94.4	91.3–99.4
Pusa 3060	70.7–81.0	81.8–86.4	91.7–98.3
Pusa 3066	57.3–87.5	86.5–91.9	92.4–97.8

**Table 2 ijms-24-16081-t002:** Agronomic performance of the PB1509-NILs carrying different resistance genes for blast and bacterial blight at IARI, New Delhi along with recurrent parent similarity.

NIL	Genotype	Genes	DF	PH (cm)	NT (cm)	PL (cm)	FG	SF (%)	TW (g)	Similarity (%)
G1	Pusa 3060-3-55-17-157-4-124-1	*Pi9* + *Pib*	83.6	102.4 *	13.3	28.2	64.0 *	75.8 *	30.2	96.83
G2	Pusa 3060-3-55-17-157-4-124-6	*Pi9* + *Pib*	84.6	104.4 *	13.3	28.4	62.3 *	79.3	33.0	96.13
G3	Pusa 3060-3-55-17-157-4-138-1	*Pi9* + *Pib*	83.3 *	101.1	12.4	29.4	50.9 *	61.9 *	32.3	96.65
G4	Pusa 3060-3-55-17-157-4-138-2	*Pi9* + *Pib*	83.0 *	102.6 *	13	28.9	64.2 *	70.3 *	33.3 *	96.60
G5	Pusa 3060-3-55-17-157-4-138-3	*Pi9* + *Pib*	83.6	98.2	13.1	29	60.5*	72.8*	32.3	97.38
G6	Pusa 3066-4-56-20-158-7-170-2	*Pita*	85.6	97.8	13.2	29.4	71.8	78.9	31.8	97.69
G7	Pusa 3066-4-56-20-158-7-171-4	*Pita*	86.3	102.4 *	12.7	30.3 *	77.6	83.2	32.6	96.87
G8	Pusa 3066-4-56-20-158-7-172-1	*Pita*	85.0	101.6	13.1	28.7	75.3	80.1	29.8	97.35
G9	Pusa 3066-4-56-20-158-7-172-2	*Pita*	84.6	103.4 *	12.4	29.1	78.2	82.6	31.4	97.23
G10	Pusa 3066-4-56-20-159-8-174-1	*Pita*	81.0 *	98.9	11.6	28.0 *	66.9 *	74.7 *	30.6	86.38
G11	Pusa 3037-1-44-3-164-20-249-2	*xa13* + *Xa21*	85.6	100.8	11.7	28.1	81.2	88.1	31.6	97.79
G12	Pusa 3037-1-44-3-164-21-255-1	*xa13* + *Xa21*	84.3	97.5	13.5	27.9	80.9	89.3	31.3	98.25
G13	Pusa 3037-1-44-3-164-21-256-4	*xa13* + *Xa21*	85.0	97.1	13	28.3	70.6 *	81.9	30.5	-
G14	Pusa 3037-1-45-5-165-22-259-4	*xa13* + *Xa21*	84.6	98.2	13.0	27.4 *	71.8	86.8	32.5	-
G15	Pusa 3054-2-47-7-166-24-261-1	*Xa38*	84.3	102.6 *	13.0	29	74.8	85.4	31.7	98.77
G16	Pusa 3054-2-47-7-166-24-261-2	*Xa38*	84.0	101.3	12.8	28.6	79.2	88.7	33.4 *	98.92
G17	Pusa 3054-2-47-7-166-24-261-3	*Xa38*	83.0	100.7	12.3	28.2	83.0	84.3	31.2	98.90
G18	Pusa 3054-2-47-7-166-24-262-2	*Xa38*	84.3	102.7 *	11.6	29.1	75.4	86.5	31.3	-
G19	Pusa 3054-2-47-7-166-24-262-3	*Xa38*	84.3	103 *	12.0	28.9	75.3	83.3	31.9	98.11
PB1509	-		85.0	98.8	12.8	29.1	78.8	83.2	31.3	-
LSD (0.05)	-		1.49	3.18	2.10	0.99	7.74	6.82	1.86	-

Where DF = days to fifty percent flowering, PH = plant height, NT = number of tillers, PL = panicle length, FG = filled grains, SF = spikelet fertility and TW = test weight. * Significantly different from PB1509 at 5% level of significance.

**Table 3 ijms-24-16081-t003:** Grain and cooking quality of the PB1509-NILs.

NIL	Genotype	HR	MR	HRR	KLBC	KBBC	KLAC	KBAC	ER	Aroma
G1	Pusa 3060-3-55-17-157-4-124-1	78	69.3	51.0	8.14	1.66	17.95	2.26	2.20	2
G2	Pusa 3060-3-55-17-157-4-124-6	78.3	70.3	51.3	8.38	1.66	17.53	2.33	2.09	2
G3	Pusa 3060-3-55-17-157-4-138-1	78.5	70	48.3	8.36	1.66	17.97	2.33	2.14	2
G4	Pusa 3060-3-55-17-157-4-138-2	79.6	70.6	50.0	8.38	1.66	16.64	2.33	1.98	2
G5	Pusa 3060-3-55-17-157-4-138-3	80.3	71.3	51.3	8.34	1.66	17.17	2.24	2.05	2
G6	Pusa 3066-4-56-20-158-7-170-2	80.3	71.6	54.0	8.23	1.66	16.95	2.33	2.05	2
G7	Pusa 3066-4-56-20-158-7-171-4	80.3	72.0 *	55.6	8.31	1.66	16.80	2.31	2.02	2
G8	Pusa 3066-4-56-20-158-7-172-1	80.6 *	71.6	50.6	8.25	1.66	16.57	2.33	2.00	2
G9	Pusa 3066-4-56-20-158-7-172-2	79.3	70.6	49.3	8.17	1.66	17.04	2.31	2.08	2
G10	Pusa 3066-4-56-20-159-8-174-1	80.0	70.3	43.6	8.03	1.66	16.37 *	2.28	2.03	2
G11	Pusa 3037-1-44-3-164-20-249-2	79.3	71.3	55.3	8.38	1.66	17.48	2.33	2.08	2
G12	Pusa 3037-1-44-3-164-21-255-1	79.6	72.0 *	54.0	8.28	1.66	18.06	2.31	2.17	2
G13	Pusa 3037-1-44-3-164-21-256-4	80.3	72.3 *	53.6	8.30	1.66	17.31	2.31	2.08	2
G14	Pusa 3037-1-45-5-165-22-259-4	80.3	72.0 *	52.0	8.42 *	1.66	18.04	2.33	2.14	2
G15	Pusa 3054-2-47-7-166-24-261-1	80.0	72.0 *	54.6	8.42 *	1.66	16.91	2.33	2.00	2
G16	Pusa 3054-2-47-7-166-24-261-2	77.6	70.0	54.0	8.28	1.66	17.71	2.31	2.13	2
G17	Pusa 3054-2-47-7-166-24-261-3	79.3	71.6	52.6	8.25	1.66	17.75	2.28	2.15	2
G18	Pusa 3054-2-47-7-166-24-262-2	78.3	70.3	52.6	8.37	1.66	18.57 *	2.31	2.21	2
G19	Pusa 3054-2-47-7-166-24-262-3	79.6	72.3 *	54.3	8.46 *	1.66	17.68	2.28	2.09	2
PB1509	-	78.6	69.8	51.1	8.21	1.66	17.56	2.31	2.14	2
LSD (0.05)	-	1.99	1.86	5.67	0.18	0.00	0.65	0.67	0.93	-

Where HR = hulling recovery, MR = milling recovery, HRR = head rice recovery, KLBC = kernel length before cooking (mm), KBBC = kernel breadth before cooking (mm), KLAC = kernel length after cooking (mm), KBAC = kernel breadth after cooking (mm) and ER = elongation ratio. * Significantly different from PB1509 at 5% level of significance.

**Table 4 ijms-24-16081-t004:** Yield performance of PB1509 NILs at three different locations namely, New Delhi (Env1), Karnal (Env2) and Urlana (Env3). * Represents the NILs with a significant difference in yield compared to RP PB1509.

NILs	Genotype	Grain Yield (Kg/ha)		
		New Delhi	Karnal	Urlana
G1	Pusa 3060-3-55-17-157-4-124-1	5008	5185	9461
G2	Pusa 3060-3-55-17-157-4-124-6	5295	5329	8245
G3	Pusa 3060-3-55-17-157-4-138-1	4322 *	4514	7410 *
G4	Pusa 3060-3-55-17-157-4-138-2	4841	3423 *	8143 *
G5	Pusa 3060-3-55-17-157-4-138-3	4914	4648	8238 *
G6	Pusa 3066-4-56-20-158-7-170-2	5221	4522	8126 *
G7	Pusa 3066-4-56-20-158-7-171-4	5559 *	4554	8514
G8	Pusa 3066-4-56-20-158-7-172-1	5524	4490	8728
G9	Pusa 3066-4-56-20-158-7-172-2	5544 *	4565	8580
G10	Pusa 3066-4-56-20-159-8-174-1	5451	4454	8893
G11	Pusa 3037-1-44-3-164-20-249-2	5712 *	5099	9709
G12	Pusa 3037-1-44-3-164-21-255-1	6267 *	4871	9744
G13	Pusa 3037-1-44-3-164-21-256-4	5749 *	4804	9021
G14	Pusa 3037-1-45-5-165-22-259-4	5513	4469	9171
G15	Pusa 3054-2-47-7-166-24-261-1	5606 *	4767	8585
G16	Pusa 3054-2-47-7-166-24-261-2	5712 *	4818	9763
G17	Pusa 3054-2-47-7-166-24-261-3	5468	5126	9415
G18	Pusa 3054-2-47-7-166-24-262-2	5781 *	4830	9131
G19	Pusa 3054-2-47-7-166-24-262-3	5372	4445	8072 *
PB1509		4995	5154	9239
LSD (0.05)		545	863	959

**Table 5 ijms-24-16081-t005:** Agronomic performance of the PB1509 NILs carrying different combinations for blast and bacterial blight resistance genes.

NIL	Genotype	Genes	DF	PH	NT	PL (cm)	TG	SF (%)	Yield (Kg/ha)
G20	Pusa 3122-27-15-165-2	*xa13 + Xa21 + Xa38*	84.5	103.5	12.5	28.0	81.5	74.1 *	8272
G21	Pusa 3122-27-16-166-1	*xa13 + Xa21 + Xa38*	85.5	106.5	14.5	27.0	75.0	75.9 *	7573
G22	Pusa 3124-37-17-168-4	*Xa38 + Pi9 + Pib*	84.5	110.5	13.5	28.0	91.0	88.6	7865
G23	Pusa 3124-37-18-175-1	*Xa38 + Pi9 + Pib*	85.0	104.0	15.5	25.0	89.0	88.3	7914
G24	Pusa 3124-38-19-176-5	*Xa38 + Pi9 + Pib*	86.5	109.5	20.5 *	31.5	92.5	93.9	8406
G25	Pusa 3124-40-21-179-4	*Xa38 + Pi9 + Pib*	88.5 *	114.5	13.5	30.0	97.5	94.8	8260
G26	Pusa 3124-40-22-180-2	*Xa38 + Pi9 + Pib*	85.5	113.0	14.0	26.5	101.0	89.0	8640
G27	Pusa 3123-33-13-312-13	*Pi9 + Pib + Pita*	84.0	113.0	14.0	29.5	91.0	84.7	8391
G28	Pusa 3123-33-13-312-22	*Pi9 + Pib + Pita*	84.0	116.0	15.5	28.5	78.5	78.8 *	7862
G29	Pusa 3123-33-13-312-25	*Pi9 + Pib + Pita*	82.5 *	110.0	12.5	27.0	89.0	93.8	8598
PB1509		*-*	85.5	105.5	14.5	28.0	89.5	90.9	8568
LSD (0.05)			1.64	10.50	5.10	4.19	18.95	8.46	1039.62

Where DF = days to fifty percent flowering, PH = plant height, NT = number of tillers, PL = panicle length, TG = total grains, SF = spikelet fertility and TW = test weight. * Significantly different from PB1509 at 5% level of significance.

**Table 6 ijms-24-16081-t006:** Grain and cooking quality of the intercrossed NILs.

NIL	Genotype	KLBC	KBBC	KLAC	KBAC	ER	Aroma
G20	Pusa 3122-27-15-165-2	8.48	1.66	17.24	2.27	2.03	2+
G21	Pusa 3122-27-16-166-1	8.08	1.66	17.76	2.33	2.19	2+
G22	Pusa 3124-37-17-168-4	8.38	1.66	17.89	2.27	2.13	2+
G23	Pusa 3124-37-18-175-1	8.63	1.66	17.89	2.33	2.07	2+
G24	Pusa 3124-38-19-176-5	8.30	1.66	17.27	2.33	2.08	2+
G25	Pusa 3124-40-21-179-4	8.66	1.66	17.29	2.33	1.99	2+
G26	Pusa 3124-40-22-180-2	8.88	1.66	17.51	2.33	1.97	2+
G27	Pusa 3123-33-13-312-13	8.61	1.66	17.51	2.33	2.03	2+
G28	Pusa 3123-33-13-312-22	8.48	1.66	17.57	2.27	2.07	2+
G29	Pusa 3123-33-13-312-25	8.41	1.66	17.87	2.33	2.12	2+
PB1509	-	8.45	1.66	17.66	2.29	2.09	2+
LSD (0.05)		0.69	0.00	0.42	0.64	0.21	-

Where KLBC = kernel length before cooking (mm), KBBC = kernel breadth before cooking (mm), KLAC = kernel length after cooking (mm), KBAC = kernel breadth after cooking (mm) and ER = elongation ratio.

**Table 7 ijms-24-16081-t007:** Blast reaction scores for the developed NILs with isolate *Mo-ei-MB20* at New Delhi and mixture of five isolates at Palampur.

NIL	Genotype	Genes	New Delhi	Palampur
G1	Pusa 3060-3-55-17-157-4-124-1	*Pi9 + Pib*	1	1
G2	Pusa 3060-3-55-17-157-4-124-6	*Pi9 + Pib*	1	1
G3	Pusa 3060-3-55-17-157-4-138-1	*Pi9 + Pib*	1	1
G4	Pusa 3060-3-55-17-157-4-138-2	*Pi9 + Pib*	2	1
G5	Pusa 3060-3-55-17-157-4-138-3	*Pi9 + Pib*	1	1
G6	Pusa 3066-4-56-20-158-7-170-2	*Pita*	2	1
G7	Pusa 3066-4-56-20-158-7-171-4	*Pita*	1	1
G8	Pusa 3066-4-56-20-158-7-172-1	*Pita*	2	1
G9	Pusa 3066-4-56-20-158-7-172-2	*Pita*	1	1
G10	Pusa 3066-4-56-20-159-8-174-1	*Pita*	2	1
G22	Pusa 3124-37-17-168-4	*Xa38 + Pi9 + Pib*	2	2
G23	Pusa 3124-37-18-175-1	*Xa38 + Pi9 + Pib*	2	2
G24	Pusa 3124-38-19-176-5	*Xa38 + Pi9 + Pib*	1	1
G25	Pusa 3124-40-21-179-4	*Xa38 + Pi9 + Pib*	2	2
G26	Pusa 3124-40-22-180-2	*Xa38 + Pi9 + Pib*	1	1
G27	Pusa 3123-33-13-312-13	*Pi9 + Pib + Pita*	0	1
G28	Pusa 3123-33-13-312-22	*Pi9 + Pib + Pita*	0	1
G29	Pusa 3123-33-13-312-25	*Pi9 + Pib + Pita*	0	1
PB 1509	-	-	5	4
P1929	-	*Pi9 + Pib*	1	1
Tetep	-	*Pita*	2	1

## Data Availability

Data are contained within the article or Appendix A.

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
