# Peer review of "Genetic Enhancement for Biotic Stress Resistance in Basmati Rice through Marker-Assisted Backcross Breeding"

_ijms, 2023, doi:10.3390/ijms242216081_

Round 1
Reviewer 1 Report
Research manuscript by Singh et al. describes the incorporation of disease tolerance alleles (bacterial blight- xa13, Xa21, Xa38; and fungal blast- Pi9, 101 Pib, Pita) into the genetic background of well-established Basmati rice variety PB1509 through marker-assisted backcross breeding (MABB). The manuscript is well-written, and the data is clear in the provided figures and Tables. This MS can be considered for publication with some changes.
Here are some suggestions:
1. Line 17-18 - bacterial blight resistance genes….; and blast resistance genes…
- Revise as-fungal blast resistance genes…
2. Line 22- SSR- write the full form of abbreviated terms while mentioning it for the first time.
3. Line 23- NILs- write the full form of abbreviated terms while mentioning it for the first time.
4. Line 42- a gram-negative bacteria…
- Revise- a gram-negative bacterium…
5. Line 40 and 44- Check the font size of reference numbers.
6. Lines 57, 65, and many other places– R genes- R should be in italics.
7. Line 60 – Xa8, xa13, Xa21 and Xa38 - and should not be in italic.
8. Line 105 – DP Full form- Donor parents? Write the full form of abbreviated terms while mentioning them for the first time.
9. Line 106 - RP Full form- recurrent parent?? Write the full form of abbreviated terms while mentioning them for the first time.
10. Line 112- RPG, full form. Write the full form of abbreviated terms while mentioning them for the first time.
11. Line 168 – RPGA full form. Write the full form of abbreviated terms while mentioning them for the first time.
12. Line 150 - DF full form. Write the full form of abbreviated terms while mentioning it for the first time.
13. Lines 161, 164, 165, 218 – HR, MR, KLBC, AEC- full forms. Write the full form of abbreviated terms while mentioning them for the first time.
14. Line 222 – Figure 3 – GGE – full form.
15. Table 5 - Write the full forms of all the abbreviations used in the table, DF, PH, ...
16. Line 245 - bacterial blight – abbreviate it to BB.
17. Figure 6 - X-axis title is missing.
18. Line 307 - Pusa 6B etc.,[22,46,47]. – Remove commas.
19. References – Numbers are mentioned twice in the whole list. Remove the duplicated numbers from all the references.
The language is clear.
Reviewer 2 Report
necessary suggestions are attached
